# Pain Biomarkers in Fibromyalgia Syndrome: Current Understanding and Future Directions

**DOI:** 10.3390/ijms241310443

**Published:** 2023-06-21

**Authors:** Martina Favretti, Cristina Iannuccelli, Manuela Di Franco

**Affiliations:** Rheumatology Unit, Department of Internal Clinical, Anesthesiologic and Cardiovascular Sciences, Sapienza University of Rome, 00161 Rome, Italy

**Keywords:** fibromyalgia, pain, biomarker, central sensitization, neuroinflammation

## Abstract

Fibromyalgia is a complex and heterogeneous clinical syndrome, mainly characterized by the presence of widespread pain, possibly associated with a variety of other symptoms. Fibromyalgia can have an extremely negative impact on the psychological, physical and social lives of people affected, sometimes causing patients to experience dramatically impaired quality of life. Nowadays, the diagnosis of fibromyalgia is still clinical, thus favoring diagnostic uncertainties and making its clear identification challenging to establish, especially in primary care centers. These difficulties lead patients to undergo innumerable clinical visits, investigations and specialist consultations, thus increasing their stress, frustration and even dissatisfaction. Unfortunately, research over the last 25 years regarding a specific biomarker for the diagnosis of fibromyalgia has been fruitless. The discovery of a reliable biomarker for fibromyalgia syndrome would be a critical step towards the early identification of this condition, not only reducing patient healthcare utilization and diagnostic test execution but also providing early intervention with guideline-based treatments. This narrative article reviews different metabolite alterations proposed as possible biomarkers for fibromyalgia, focusing on their associations with clinical evidence of pain, and highlights some new, promising areas of research in this context. Nevertheless, none of the analyzed metabolites emerge as sufficiently reliable to be validated as a diagnostic biomarker. Given the complexity of this syndrome, in the future, a panel of biomarkers, including subtype-specific biomarkers, could be considered as an interesting alternative research area.

## 1. Introduction

Fibromyalgia (FM) is a chronic clinical syndrome characterized by widespread pain (WP), fatigue and sleep disturbances. In addition, neuropsychiatric manifestations—such as cognitive impairment and mood disturbances—and somatic and dysautonomic symptoms can be present. The global prevalence of FM is estimated at 2.7%, with a higher frequency in women, patients over 50 years of age, those with a lower socio-economical level and those with obesity [1]. The accurate prevalence of FM is difficult to determine, considering the increasing evidence of frequent misdiagnosis, facilitated by the lack of reliable diagnostic biomarkers; furthermore, FM is rarely an isolated syndrome and its comorbidity with other chronic pain conditions or mental health disorders is frequent, leading to an increased likelihood of misdiagnosis [2,3]. The lack of reliable biomarkers is a central problem and barrier in FM diagnosis, not only because it is associated with an increased risk of misdiagnosis but also because it makes it difficult, if not impossible, to identify this syndrome in the earlier stages [4]. Moreover, diagnostic uncertainties linked to the absence of biomarkers increase patients’ frustration and dissatisfaction. These patients frequently seek even more medical attention and undergo numerous investigations and specialist consultations, favoring a rise in direct and indirect healthcare costs [5]. Since any serological, imaging or histological marker is currently available, the diagnosis of FM is still exclusively clinical; however, classification criteria may be used to confirm the diagnosis. The first set of classification criteria was developed in 1990 by the American College of Rheumatology (ACR) and required a history of WP and tenderness in 11 of 18 tender points [6]. In 2010, the ACR published its preliminary diagnostic criteria for FM [7], with the aim to overcome some practical and philosophical problems of the 1990 criteria; according to these criteria, a diagnosis of FM can be made in the presence of WP, evaluated by the Widespread Pain Index (WPI), and associated typical symptoms, assessed by the Symptom Severity Scale (SSS). These criteria were subsequently revised in 2016, adding the generalized pain criterion and stating that the diagnosis of FM does not exclude the presence of other clinically important illnesses [8]. Regardless of the differences between these criteria, one main feature remains unchanged: the presence of WP. Pain is a central symptom in FM; it is typically chronic, as it persists or recurs for longer than three months [9], and generalized, as it is localized in at least four out of five body regions [8]. It is associated with significant emotional distress and functional disability and cannot be better explained by other chronic pain conditions. According to the new International Association for the Study of Pain (IASP) classification of chronic pain [10], these features enable us to identify FM as a chronic primary pain (CPP) condition. CPP is a new diagnosis developed with the aim to define conditions characterized by a complex interaction of biological, psychological and social factors and therefore overcome the obsolete dichotomy of ‘physical’ and ‘psychological’ pain. Different chronic pain states fall within this new definition and at least some of them—including FM—can be defined using the pathophysiological descriptor of ‘nociplastic pain’ [11]. This new term refers to ‘pain that arises from altered nociception despite no clear evidence of actual or threatened tissue damage, causing the activation of peripheral nociceptors or evidence for disease or lesion of the somatosensory causing the pain’, and it is likely provoked by changes in nociceptive processing, probably due to central nervous system (CNS) activity modifications [11,12]. Nociceptive processing derives from the transmission of a stimulus from peripheral tissue to the brain and can be modulated in different ways. Changes in transmission mechanisms have been shown using different experimental pain tests and new imaging techniques such as functional magnetic resonance imaging (fMRI), proton spectroscopy or positron emission tomography (PET) [13]. Furthermore, these modifications of nociceptive perception, transduction and transmission could also be identified by the level modification of different metabolites. The aim of this narrative review was to collect metabolites likely involved in chronic pain perception whose alteration has been described in FM patients, in order to identify possible biomarkers of pain in FM syndrome.

## 2. Pathophysiology of Pain in FM Syndrome

Despite extensive studies, the exact pathogenesis of FM remains not fully understood, but different risk factors and pathophysiological mechanisms, potentially implicated in the development of the disease, have been identified. Pain in FM can be either described as dull, deep or aching—traditional descriptors of nociceptive pain—or burning or tingling—usually defining neuropathic pain [13]. However, as a hallmark of nociplastic pain, no clear evidence of primary tissue damage exists. In 2013, Üçeyler N. et al. [14] described the presence of small fiber neuropathy (SFN) in FM patients, supporting the hypothesis of small fiber damage as the first trigger of pain in FM syndrome. Since then, different studies have been published either supporting [15,16,17] or disproving [18,19] this hypothesis. The exact role of SFN in the pathogenesis of FM remains unclear. Thus far, it is assumed that at least some FM patients can develop SFN as a consequence of the same pathophysiological mechanism underlying the syndrome, but SFN cannot be considered the first pathogenetic event [20]. Typically, FM patients display enhanced sensitivity to painful and nonpainful stimuli, secondary to the development of hyperalgesia, allodynia or the abnormal wind-up of secondary pain [21]. These and other distinctive features are the consequences of functional and morphological changes in the CNS, primarily in pain processing brain structures [22]. Central sensitization (CS) is supposed to be the pathophysiological mechanism underlying these changes, possibly supported by glial activation due to neuroinflammation triggers. Moreover, additional aberrant systems could contribute to enhanced pain in FM patients: these include autonomic nervous system (ANS) abnormalities and hypothalamic–pituitary–adrenal (HPA) axis dysfunction [23].

### 2.1. Central Sensitization

According to the IASP, CS can be defined as the ‘increased responsiveness of nociceptive neurons in CNS to their normal or subthreshold afferent input’ [20]. CS is characterized by evidence of dysfunctional descending inhibitory pathways and increased facilitative activity, resulting in temporal summation with enlarged receptive fields, lower nociceptor thresholds, increased spontaneous neuronal activity and augmented stimulus responses [24]. Physiological nociceptive transmission strictly depends on the balance of excitatory and inhibitory stimuli [25]. The ascending system starts with primary afferent neurons—myelinated Aδ-fibers and unmyelinated C-fibers—conveying the noxious stimulus to the projection neurons in the dorsal horn (DH) of the spinal cord, a central network of integrating inputs [26]. Subsequently, these projection neurons transmit the noxious information either to the somatosensory cortex or to the cingulate and insular cortices, mainly via the spinothalamic tract [25]. A crucial nucleus of the descending system is the periaqueductal grey (PAG), which integrates information from higher structures of the brain—including the hypothalamus, amygdala and frontal lobe—and ascending nociceptive impulses and transmits inhibitory stimuli to the DH of the spinal cord [27]. Primary afferent fiber activation is physiologically associated with glutamate (Glu) release and the short post-synaptic depolarization of second neurons via alpha amino-3-hydroxy-5-methyl-4-isoxazolepropionic acid receptor (AMPA-R) activation. Indeed, the repetitive stimulation and co-release of other neuropeptides, such as substance P (SP), induce sustained post-synaptic depolarization with subsequent N-methyl-D-aspartate receptor (NMDA-R) unlock, due to Mg^2+^ removal [28]. NMDA-R activation results in Ca^2+^ influx and hyperexcitability of the DH neurons, favoring CS. In turn, hyperexcitability of the DH neurons facilitates NMDA-R activation, sustaining CS persistence [29]. Therefore, CS is the result of repetitive stimulation of nociceptors that activate several mechanisms of neuroplasticity, such as the reduction of pain thresholds, amplification of pain responses and spread of pain sensitivity to non-injured areas, leading to the enhanced excitability of the neurons of the DH of the spinal cord [30]. Both structural and functional modifications of the cortical regions implicated in pain processes and imbalanced levels of neurotransmitters involved in pain sensation have been associated with CS.

Several studies in FM patients have demonstrated that similar alterations in imaging and neurotransmitters could be present, corroborating the hypothesis of pain as a consequence of CS. Different noninvasive neuroimaging methods are currently available to enable a better understanding of pain processes. These methods include structural imaging techniques, such as voxel-based morphometry (VBM), and functional imaging techniques, such as fMRI and resting-state functional magnetic resonance imaging (rs-fMRI). A recent meta-analysis of VMB studies comparing CPP patients with healthy controls (HC) identified grey matter alterations in pain-processing brain regions, such as the cingulate, prefrontal and insular cortices, in patients affected by CPP [31]. Most of the studies conducted in FM patients indicate similar regional atrophies, probably secondary to chronic pain development, but none of the structural changes are specific to FM syndrome [20]. Conversely, different functional changes related to CS in FM patients have been described. Studies comparing noxious stimuli responses using fMRI demonstrate that, when an equal-pain-intensity stimulus is applied, FM patients display similar but more extensive patterns of brain activation than HC, particularly observed in the posterior insula and secondary somatosensory cortex [32,33]. Furthermore, fMRI studies of descending inhibitory pathways have identified modified PAG connectivity in FM patients, possibly associated with decreased activation of the anti-nociceptive system [20,34]. These findings are consistent with the hypothesis that a deficit in pain inhibition systems is specifically involved in chronic pain development in FM syndrome. Two principal descending inhibitory pathways exist: the opioidergic and 5-hydroxytryptaminergic-noradrenergic pathways. Endogenous opioid tone appears to be normal or even increased in FM patients [35], while endogenous serotoninergic and noradrenergic activity is apparently decreased, leading to reduced conditioned pain modulation (CPM) [36]. Finally, rs-fMRI studies in FM patients showed imbalanced resting-state connectivity in pain-processing regions with greater neuronal activation, suggesting that chronic pain could induce changes in brain processes activated even in the absence of an external stimulus [32,37].

### 2.2. Neuroinflammation

Neuroinflammation is a form of localized inflammation occurring in the peripheral nervous system (PNS) and CNS and it is characterized by vascular changes with increased permeability, glial cell activation, the infiltration and activation of leukocytes and the increased production of inflammatory mediators including cytokines and chemokines [38]. While acute inflammation can elicit acute pain sensations, neuroinflammation is supposed to be implicated in the chronification and persistence of pain. Compelling evidence exists proving that cytokines, chemokines and other glia-produced mediators could either induce or maintain CS, driving the development of WP [39]. Apparently, neuroinflammation is the result of a bidirectional crosstalk between nociceptors and different non-neuronal cells, particularly immune and glial cells. Cytokines and inflammatory mediators provided by immune and glial cells activate and sensitize nociceptors, while nociceptors themselves can secrete cytokines and chemokines, which are essential for immune modulation [39].

Regarding immune cells, mast cells (MCs) seem to play a central role in neuroinflammation development. MCs are resident immune cells located ubiquitously in vascularized tissue, predominantly at the interface with the external environment. These cells can be found close to small-caliber sensory nerve fibers in peripheral innervated tissue, on the endoneurial compartments of peripheral nerves and, in small quantities, in the brain [40]. In the brain, MCs can be found on the abluminal sides of blood vessels, where they interact with neurons and glial cells [41]. Different receptors—such as toll-like receptor (TLR) 2 [42], TLR4 [42] or P2X receptors [43]—induce MC degranulation with the consequent release of various mediators, such as biogenic amines, cytokines, neuropeptides, growth factors or ATP [42]. Mediators secreted by MCs promote nociceptor activation and hypersensitization, both directly—via pro-nociceptor mediators’ production [44]—and indirectly—inducing neuropeptide release [42]. Furthermore, there is substantial evidence that MCs can impact blood–brain barrier (BBB) permeability, disrupting its integrity [45]. Therefore, in the absence of tight control, MC–nociceptor interactions induce nociceptive receptors’ hyperactivity while lowering pain thresholds. Persistent nociceptive stimulation finally sensitizes the neurons of the DH, leading to CS [44]. Glial cells are CNS cells that, in addition to MCs, seem to play a role in neuroinflammation and the chronification of pain. Among these cells, microglia and astrocytes’ involvement is better understood, but there is evidence of an active and context-specific role of oligodendrocytes as well [38]. Local microglia represent the resident macrophage population of CNS and are able to undergo rapid activation in response to minor pathological changes in the CNS. Different mediators released by immune cells and nociceptors—such as adenosine triphosphate (ATP) and chemokine ligand (CXCL) 1—could trigger microglia’s production of other mediators, either cytokines—such as tumor necrosis factor (TNF), interleukin (IL) 1 and IL-18—or growth factors—such as brain-derived neurotrophic factor (BDNF) [46]. These mediators are consequently able to enhance spinal pain transmission. Neuronal hyperexcitability and augmented pain signal transmission are also induced by astrocyte activation. In the context of neuroinflammation, there is evidence that astrocytes lose their ability to maintain the homeostatic concentrations of Glu and K^+^ while achieving the capacity to secrete ATP, Glu and chemokines, thus contributing to chronic pain development [47]. The activation of microglia and astrocytes has been described in different neurodegenerative diseases [48,49] and neuropsychiatric illnesses [50,51] using imaging techniques, such as PET and proton magnetic resonance spectroscopy (^1^H-MRS) [52], but it is supposed to be present in FM syndrome too.

A recent study [53] described elevated choline levels, a marker of glial activation, in the anterior insula of FM patients compared to HC, and reduced connectivity between the anterior insula and putamen. Both these findings were associated with worse pain interference in FM patients. A previous study [54] showed higher choline levels in the right dorsolateral prefrontal cortex (PFC) in FM patients, also describing a positive correlation with pain. According to these findings, but also on the basis of evidence of neuropeptide and cytokine alterations in FM patients, it has been proposed that neuroinflammation, or at least neurogenic inflammation, could be involved in the development and maintenance of pain in FM syndrome. Neurogenic inflammation is the consequence of nociceptor activation—particularly C fibers—and the associated release of neuropeptides such as SP, calcitonin gene-related peptide (CGRP) and prostanoids [55]. This augmented neuronal activity could possibly trigger neuroinflammation in peripheral tissue [56].

### 2.3. Neuroendocrine and Autonomic Nervous System Dysfunction

Stress stimuli, which can be either real or perceived and arise from different life events, are known modulators of the pain system. Under physiological conditions, exposure to a stressor elicits the rapid activation of the ANS and the HPA axis, with the consequent release of catecholamines and cortisol, starting the so-called ‘fight or flight’ response [57]. The induction of anti-nociceptive and analgesic mechanisms is a hallmark of this response [58]. Otherwise, deregulation of the stress response, caused by exaggerated or maladaptive reactions, is associated with altered nociceptive responses, eventually leading to chronic pain development. Usually, in response to a stressor stimulus, the corticolimbic system, mostly represented by the prefrontal cortex, amygdala and hippocampus, sends disinhibitory impulses to the hypothalamus. The first quasi-instantaneous response to hypothalamus disinhibition is catecholamine release—both noradrenaline and adrenaline—via locus coeruleus stimulation. Catecholamines induce a hypervigilant state that is characterized, among other features, by augmented pain resistance secondary to descending inhibitory pathway activation [59]. Conversely, pathological hyperactivity of the sympathetic system is associated with primary afferent nociceptor activation and, subsequently, the development of chronic pain [60]. Hypothalamus activation results also in corticotropin releasing hormone (CRH) secretion, which induces the pituitary gland release of adrenocorticotropic hormone (ACTH), which, in turn, activates the cortical adrenal gland’s production of cortisol. Cortisol stimulates immediate—on the scale of minutes—and tardive—due to structural and genomic effects—systemic responses. The corticolimbic system and hypothalamus show high affinity for cortisol and, therefore, its production controls the stress response via a negative feedback loop [61]. There is evidence that chronic stress induces functional and morphological changes in key regulatory regions of the stress response, especially the corticolimbic system [62]. These changes are associated with numerous abnormalities, including an unbalanced ANS and dysfunctional HPA axis, which have been described in various chronic pain conditions. Interestingly, it seems that distinct chronic pain conditions arise from a complex and partly unique interaction not only between steroid production but also between the HPA axis, ANS and immune system, possibly explaining the pathophysiological and clinical discrepancies between them [63].

Regarding FM, hyporeactivity of the HPA axis, often associated with lower basal cortisol levels, and basal hyperactive sympathetic activity have been described. Despite discrepancies between studies, FM patients present plasma cortisol level alterations characterized by the flattening of the cortisol concentration during the day, due to the generally dysregulated circadian variation of this hormone [33]. Furthermore, the administration of different stress stimuli, such as IL-6 injection or hypoglycemia induction, has been associated with a delayed or reduced ACTH response, showing the hyporeactivity state of the HPA axis [64]. ANS activity has been investigated using power spectrum analysis of heart rate variability (HRV). These studies described reduced HRV in FM patients, as a consequence of hyperactivity of the sympathetic nervous system and decreased parasympathetic tone [60,63]. As well as dysregulation of the neuroendocrine axis, which is widely described in FM syndrome, different physical traumas and psychological triggers are associated with enhanced pain in FM patients [23]. Several types of stressors have been proposed as potential environmental triggers of the disease, possibly influencing gene expression and leading to HPA axis impairment [65]. These potential stressors include adverse events in neonatal or childhood life [66] or repeated physical or psychological stressors [23]. Environmental factors are believed to promote FM in genetically predisposed individuals, and the genetic predisposition is supported by evidence of strong familial aggregation among FM patients [67]. Single-nucleotide polymorphisms (SNPs) of several genes have been associated with FM and, among these, the catechol-O-methyl transferase (COMT) gene is the most widely investigated [68]. COMT is the major enzyme involved in monoamine metabolism, such as dopamine, adrenaline and noradrenaline, and alterations in COMT activity are associated with chronic pain development [69]. COMT gene polymorphism has been considered a possible predisposing factor for the development of FM but current reports remain contradictory [70,71].

## 3. Biomarkers of Pain in Fibromyalgia Syndrome

### 3.1. Glutamate

Glutamate (Glu) is the principal excitatory neurotransmitter of the nervous system, and it plays a significant role in nociceptive modulation: it regulates other neurotransmitters’ release from afferent nerve fibers, stimulates the activation of DH neurons and its neuroplasticity and induces cortical activation, facilitating the storage and generation of long-term memory processes [25]. Dysfunction in Glu and other neurotransmitters’ metabolism and the consequently altered availability have been implicated in different chronic pain conditions, especially through ^1^H-MRI studies. Interestingly, the results of these studies suggest the existence of a unique neurometabolite signature for each different pain condition [72].

FM patients demonstrate a general increase in Glu levels, possibly playing a significant role in the development and maintenance of chronic pain [73]. FM patients also demonstrate higher cerebrospinal fluid (CSF) Glu levels than chronic migraine patients [74], and HC and Glu levels positively correlate with neurotrophin production but not with clinical evidence of pain [75]. These findings suggest that, despite the difficulty to relate Glu levels to clinical evidence of CS, their involvement in FM syndrome is unquestionable [75]. Subsequently, ^1^H-MRI studies have revealed significantly higher Glu levels in FM patients compared to HC in different brain regions, such as the posterior insular cortex [76,77], posterior cingulate gyrus [78], right amygdala [79] and ventrolateral PFC [80] Table 1. These findings lead to the hypothesis that Glu hyperactivity in pain-processing brain regions—due to more Glu in synaptic vesicles, a higher number or density of glutamatergic synapses or less Glu reuptake—causes FM syndrome [76]. There is strong evidence that Glu excess can lead to neuronal dysfunction and cell loss, possibly explaining the brain regional atrophy described in FM patients [81]. An alternative hypothesis states that neuronal dysfunction could be the consequence of astrocyte deficits. Astrocytes are implicated in the uptake, metabolism and recycling of Glu and their alteration could start a cascade of metabolic events leading to abnormal neurotransmission [78]. Additionally, there is evidence that Glu levels in particular brain regions correlate with clinical pain experience in FM patients, even when these levels are not significantly higher than in HC. Significant correlations have been identified in the posterior cingulate gyrus [78], left thalamus [80] and insular cortex [76]. Regarding the insular cortex, Lee J. et al. [82] demonstrated that the anterior and posterior insular cortices’ Glu levels are associated with different dimensions of chronic pain: the cognitive affective and sensory physical dimensions, respectively. These findings are in line with the different roles that these two brain regions play in pain processing [83]. Furthermore, Harris R.E. et al. [77] demonstrated that changing levels of insular Glu correlate with changes in clinical pain experience in FM patients.

This evidence suggests that Glu level detection in particular brain regions, assessed by neuroimaging techniques, could be a useful biomarker of pain in FM patients. Unfortunately, several limitations exist. First of all, it is unlikely that Glu measurements reflect the synaptic levels of this neurotransmitter: not only is the ^1^H-MRS Glu signal often contaminated by glutamine levels, but, arising from grey and white matter signals, it ensembles multiple different cell types, including regions distant from synapses [84]. Secondly, there are no standardized ^1^H-MRS parameters for neurometabolite level estimation and there is little consensus on the best way to measure Glu levels [72]. Therefore, the assessment of CNS Glu levels as a biomarker of pain using ^1^H-MRS nowadays remains scarcely affordable and poorly reproducible.

### 3.2. Substance P

Substance P (SP) is a neuropeptide member of the tachykinin family, along with neurokinin (NK) A and NKB. SP is expressed throughout the nervous and immune systems, and it is involved not only in a large number of physiological processes but also in numerous pathological conditions [85]. The biological SP actions are mediated by neurokinin (NK) receptors. Three NK receptors exist—NK_1_-R, NK_2_-R and NK_3_-R—and although each receptor has moderate affinity for each tachykinin ligand, every single NK receptor is preferentially activated by one ligand. NK_1_-R is the primary receptor for SP [86]. SP plays a central role in pain transmission and compelling evidence suggests that it could be implicated in the induction and maintenance of CS, as well as the activation of neurogenic inflammation. Primary afferent sensory fibers express SP along with Glu and are able to release this neuropeptide both centrally in the DH neurons’ synapses and backward in the peripheral terminals [87]. SP released in the DH of the spinal cord binds NK_1_-R and induces the hyperexcitability of the spinal neurons, both directly by causing a slow excitatory postsynaptic potential and indirectly by facilitating the activation of NMDA-R, thus favoring initial CS development [88]. Spinal neurons’ hyperexcitability is then maintained by SP release. NK_1_-R activation is able to increase Ca^2+^ influx, subsequently activating C-fos and C-jun, two proto-oncogenes involved in the persistence of sensitization [89]. Additionally, there is evidence that CGRP, released together with SP, enhances NK_1_-R gene expression, thus favoring its activation and, therefore, CS persistence [88]. By contrast, SP released in the peripheral endings of primary sensory neurons induces arteriolar dilatation, plasma extravasation and leukocyte infiltration, thus generating neurogenic inflammation [90]. Green D.P. et al. [91] demonstrated that these effects are mediated by MrgprX2 activation, a receptor expressed exclusively on MCs’ surfaces.

Considering SP’s involvement in pain processes, it has been suggested that this neuropeptide could be implicated in the pathogenesis of FM syndrome, and the measurement of SP levels—either from CSF or peripheral samples—has been proposed as a possible diagnostic biomarker (Table 2). Different studies [92,93,94] demonstrate significantly higher SP levels in the CSF of FM patients compared to HC, but no relation [93] or, surprisingly, an inverse relation [92] with the clinical severity of pain was found. One study [95] did not report any differences in SP serum levels between FM patients and HC. Conversely, two more recent studies [96,97] demonstrated higher serum levels of SP in FM patients compared to HC. None of these studies compare serum SP levels with clinical pain scores. It has also been proposed that the beneficial effects of different complementary FM treatments could be explained by SP level changes after treatment administration. It has been shown that there is a significant SP serum level reduction after the administration of massage therapy [98], acupuncture [99] and dietary supplements containing primarily an extract of salmon’s milt [97]. All of these treatments have been demonstrated to improve the clinical evidence of pain but a direct correlation with SP levels has not been found. A previous study [100] showed an increase in SP serum levels after acupuncture treatment, despite a significant reduction in pain scores. However, unlike the other studies, this was not a randomized controlled trial. Another study [101] investigated the effect of different exercise and walking programs on pain and SP level changes in the CSF of FM patients, demonstrating a significant association between changes in pain threshold and changes in SP levels after exercise. According to these findings, it appears sufficiently clear that abnormal SP levels can be found in FM patients, both in CSF and peripheral samples, but the actual association of this marker with the clinical evidence of pain remains less clear.

### 3.3. Nerve Growth Factor

Nerve growth factor (NGF) is a member of the neurotrophin family and participates in the survival and growth of distinct neuronal populations, especially during development. In contrast, during adulthood, NGF changes its role and is involved in modulating nociceptive transmission [89]. Neurotrophins can bind two types of receptors: one is the tyrosine-kinase (Trk) receptor, which is highly selective—NGF binds preferentially to TrkA—and the other, named p75^NTR^, is much more promiscuous [102]. It has been demonstrated that the exogenous administration of NGF is able to induce both thermal—early onset—and mechanical—delayed—hyperalgesia in healthy volunteers, lasting up to 7 days [103,104,105], with pain sensations expanding both proximally and distally after NGF injections [106,107,108]. These findings suggest NGF’s ability to induce both peripheral and central sensitization. Rapidly after stimulation, TrkA expressed on the peripheral terminals of nociceptive fibers reduces the nociceptor threshold by inducing the sensitization of different ion channels, thus favoring early-onset peripheral sensitization [109]. Subsequently, the NGF/TrkA complex is retrogradely transported to the neuronal soma in the dorsal root ganglia (DRG), where it stimulates neuropeptide release—such as SP, CGRP and BDNF—either in peripheral or central terminals, and upregulates the gene expression of both ion channels and neuropeptides. These delayed effects contribute both to peripheral and central sensitization [102]. Finally, a third mechanism could be implicated in NGF-mediated sensitization: it might cause pain by promoting the sprouting of nociceptive neurons. There is evidence of pathological peripheral nerve sprouting and hyperinnervation in disease models of bone cancer pain [110] and it has been hypothesized that a similar effect could be induced also in central sites such as the DH of the spinal cord [111]. The increased fiber density could stimulate increased nociceptor responsiveness, leading to augmented pain transmission.

NGF appears to be little expressed in normal adult conditions but there is evidence of augmented NGF levels in multiple pathological states, especially those associated with inflammation. For this reason, it has been proposed that NGF could perform an immunomodulatory function in inflammatory response regulation, including MC differentiation, maturation and degranulation [112]. However, this feature remains controversial: although there is evidence of a possible NGF proinflammatory action on MCs [113,114], several recent reports demonstrate that NGF does not affect MC activation [115] and contradictory evidence exists regarding NGF receptor expression on MC surfaces [102]. In contrast with peripheral production, NGF expression in the CNS appears to be much more limited. In the CNS, NGF can be found primarily in the basal forebrain cholinergic neurons of limbic areas. The presence of NGF—and BDNF—in areas involved in neuroendocrine response regulation indicates that these neurotrophins could be involved in the modulation of the endocrine response to stress, and neurotrophin-mediated neuronal plasticity could increase the susceptibility to stress-related psychiatric disorders [116].

Considering the putative role of NGF in chronic pain, and according to the evidence of altered levels of NGF in different chronic pain conditions [117,118], it has been hypothesized that abnormal NGF levels could be observed in FM patients and contribute to the pain experience (Table 3). However, only two studies have measured NGF levels in the CSF of FM patients, and both detected significantly higher NGF levels in FM compared to HC [75,119]. Sarchielli P. et al. [75] also demonstrated a positive correlation between NGF levels and BDNF levels and both correlated with the duration of chronic pain but not with clinical evidence of pain. This evidence suggests that NGF favors CS in FM patients via BDNF upregulation, thus sustaining pain persistence. Two additional studies evaluated NGF levels from peripheral samples of FM patients, with conflicting results. Baumeister D. et al. [120] did not find differences in NGF levels between FM patients and HC, refuting the notion that peripheral growth factor levels contribute to the pathophysiology of FM syndrome and peripheral sensitization development. Jablochkova A. et al. [121] showed reduced peripheral NGF levels in FM patients compared to HC but did not find any relation with other serological biomarkers or clinical data. According to the authors, this finding could be explained by the emotional status of patients, considering that major depression disorder has been associated with low serum NGF levels. Unfortunately, the study did not demonstrate a significant relation of NGF levels with depressive symptoms. Furthermore, the authors speculatively proposed that low peripheral levels of NGF could explain the nociceptive fiber alterations described in FM patients, such as the reduction in distal intraepidermal nerve fiber density or C-fibers with smaller diameters. Nonetheless, considering the paucity and inconsistency of the current literature, it is difficult to consider NGF levels as a reliable biomarker of pain in FM syndrome.

### 3.4. Brain-Derived Neurotrophic Factor

BDNF is another member of the neurotrophin family and it takes part in the regulation of different processes, such as the development of brain circuits and the formation and maintenance of the neuronal morphology or brain architecture [122]. Moreover, it is well established that the altered expression of BDNF can initiate and maintain inflammatory, neuropathic and chronic pain, by mediating pain plasticity and sensitization processes [123]. Moreover, an anti-nociceptive role of BDNF has been occasionally described, suggesting that it can act in different ways depending on the specific intracellular pathway activated and the different cell types involved [124]. BDNF is expressed by different neuronal and non-neuronal cells. Among neuronal cells, BDNF expression has been demonstrated in small- and medium-sized sensory neurons [125] but also in several brain areas, such as the hippocampus, cortex, amygdala, striatum and hypothalamus [126]. Along with neuronal cells, BDNF is also synthetized by immune system cells, especially microglial cells, and platelets [127]. It has been demonstrated that various stimuli are able to induce BDNF release in different cells. Distinct electrical activity patterns, including Glu-mediated stimuli, are efficient to induce BDNF release in the CNS to mediate synaptic as well as network plasticity processes [126]. In the PNS, different pathological conditions associated with the enhanced retrograde transport of NGF to the DRG increase BDNF gene expression and central terminal release, thus favoring sensitization processes [128]. Lastly, various extracellular stimuli can induce the microglial release of BDNF and, among them, P2X4R activation, mediated by extracellular ATP, has been associated with central sensitization features such as pain hypersensitivity or allodynia [129]. BDNF’s activities are mediated by high-affinity TrkB and low-affinity p75^NTR^ receptor binding [102]. BDNF released from nociceptors in the DH of the spinal cord binds TrkB receptors expressed by medium- to large-sized neurons in the spinal cord, including ascending projection neurons, and induces the pre- and post-synaptic potentiation of Glu transmission via NMDA-R plasticity [124]. Moreover, the spinal microglial secretion of BDNF binds neuronal TrkB receptors and contributes to pain hypersensitivity through the disinhibition of pain processes: it suppresses the intrinsic inhibitory circuits in the DH by decreasing the expression of potassium-chloride cotransporter 2 (KCC2) and weakening GABAergic inhibitory synapses [127]. BDNF is also a central factor in long-term potentiation (LTP) processes, a core mechanism of plasticity in the nervous system, and it has been shown to be sufficient to induce LTP in the DH neurons, thereby maintaining CS [130].

A potential role of BDNF in the pathophysiology of FM and the development of pain have been extensively studied in the last few years, with various results. Several studies demonstrated higher CSF [75], plasma [121,131] and serum [132,133,134] BDNF levels in FM patients compared to HC. These findings advocate for a key role of BDNF in FM pathophysiology, especially as a pain modulator. Given the lack of relation between serum BDNF levels and illness duration, Laske C. et al. [132] concluded that the increased BDNF serum concentration is rather due to peripheral pain modulation. Subsequently, Nugraha B. et al. [135] suggested that higher serum BDNF levels in FM patients may be part of an adaptation mechanism in chronic pain, or, considering its possible anti-nociceptive role, a long-term high concentration of BDNF may act as a defensive mechanism. However, the absence of differences in plasma [136] and serum [120,137] BDNF levels between FM patients and HC have also been described. Moreover, a study by Iannuccelli C. et al. [138] described significantly lower BDNF serum levels in FM patients compared to HC. Different explanations could be made for these findings. First, the heterogeneous nature of FM makes it difficult to generalize, and the mechanisms underlying CS are difficult to establish [120]. Secondly, although it has been demonstrated that circulating BDNF represents 70–80% of that produced in the CNS [139], BDNF levels could be influenced by other cellular sources, such as platelet degranulation in clotting processes—which could influence especially plasma BDNF levels [140]—or skeletal muscle production in response to contraction stimulation [141]. Lastly, it must be taken into account that the chronic administration of antidepressants modifies BDNF expression [142] and could therefore interfere with the levels of this mediator. Bidar A. et al. [143] actually showed a rapid reduction in serum BDNF levels in FM patients after one-month treatment with duloxetine, thus supporting the overall role of this metabolite in pain modulation. Since this study did not show any significant difference in BDNF levels between FM patients and patients affected by non-FM chronic nociceptive pain disorders, the authors also noted that their finding weakens the exclusive role of BDNF in nociplastic pain. Nonetheless, even studies that have described significantly higher BDNF levels in FM patients are not able to demonstrate any relation with clinical pain [75,121,131,132,134]. Only one study [144] demonstrated a significant inverse relation between BDNF levels and pressure pain thresholds, supporting its value as a potential biomarker of pain in FM patients. Recently, a possible association between BDNF levels and dysfunction in the descending pain modulatory system in FM syndrome has been studied. Caumo W. et al. [145] demonstrated that central sensitization syndromes (CSS) without a structural pathology, including FM, present higher BDNF serum levels than in patients affected by CSS with a structural pathology, such as osteoarthritis. The authors were also able to show an inverse relation between BDNF serum levels and CPM, favoring a higher propensity for pain. In addition, Soldatelli M.D. et al. [146] described an association between dysfunction of the descending modulatory system and the severity of FM symptoms related to psychological aspects, including pain catastrophizing and disability due to pain.

It is therefore reasonable to conclude that altered BDNF production is implicated in the pathophysiology of FM, most likely regulating neuroplasticity processes that favor CS (Table 4). On the other hand, BDNF’s association with clinical evidence of pain remains controversial. Further studies are needed not only to better investigate BDNF’s role in the development of FM and pain but also to clarify the most reliable sample, between serum and plasma, to measure BDNF levels of CNS origin.

### 3.5. Mu Opioid Receptor

Dysregulation of the descending anti-nociceptive system is a potential pathophysiological feature of chronic pain in FM syndrome. Compelling evidence regarding decreased endogenous noradrenergic and serotoninergic activity in FM syndrome exists [147], while the endogenous opioid system appears to affect CPM in a different way. It has been largely demonstrated that opioid treatment in FM patients is not effective [148,149], while there is promising evidence of low-dose opioid antagonism’s efficacy in the management of FM symptoms [150,151,152]. Endogenous enkephalin levels appear to be higher in the CSF of FM patients [35], while Mu opioid receptor (MOR) availability is reduced in different pain-processing brain regions [153,154,155] and shows a significant relation with the affective dimension of pain [153]. Authors have suggested that the reduced MOR availability in FM patients could be the consequence of an overactive opioidergic system, which could lead to MOR’s reduced affinity or the downregulation of GABAergic inhibitory interneurons [153]. As a result of the reduced MOR availability, phasic endogenous opioid release occurring during noxious stimulation is no longer able to produce the appropriate inhibition of GABAergic interneurons, leading to the development of pain [154].

Recently, Üçeyler N. et al. [155] proposed a possible neuroimmune connection linking IL-4 gene expression to cerebral opioid receptor availability in FM patients. Endogenous opioid system and immune system cells are known to engage in bidirectional and complex communication [156], and it has been proposed that opioid receptors, including MOR, expressed by immune cells could be implicated in pain modulation [157,158]. Two recently published studies demonstrated a lower percentage of B cells [159] and natural killer cells [160] expressing MOR in FM and osteoarthritis patients with a moderate/severe intensity of pain compared to HC, and an inverse relation between the percentage of MOR B lymphocytes and natural killer cells and pain intensity. The authors proposed the determination of the percentage of MOR cell expression as a possible biomarker of pain in FM syndrome but also in other chronic pain states. Altogether, these studies indicate MOR’s reduced expression as an interesting possible biomarker of pain in FM syndrome. However, further studies are needed, not only to confirm the possible use of this marker but also to investigate the potential relation between MOR availability in brain regions and peripheral immune cells’ MOR expression, in order to deepen our knowledge regarding endogenous opioid system dysfunction in FM.

### 3.6. Mast Cells and Cytokine Production

Several FM symptoms—such as hyperalgesia, sleepiness and fatigue—are remarkably similar to those that occur after contact with infectious agents, with the consequent stimulation of inflammatory cytokines, and accumulating evidence supports the hypothesis that FM patients could present the dysfunctional regulation of cytokine production [161]. FM symptoms can be worsened by stress, and stressful stimuli are able to trigger MC degranulation and the release of proinflammatory and neuro-sensitizing mediators activating both nociceptors and microglial cells [162]. As well as stressful stimuli, mediators released by nociceptive sensory neurons can regulate the maturation, recruitment and degranulation of MCs [40]. Starting from this assumption, it has been hypothesized that dysfunctional MCs and the associated altered cytokine production could be implicated in the pathogenesis of pain in FM. MCs appear to be overexpressed in the papillary dermis in FM patients [163,164]. It has been suggested that excessive SP release could trigger MC activation [163], and increased numbers of MCs’ dermal mediators could represent a source of repetitive peripheral stimuli reaching the CNS, which cause CS [164]. However, a clinical trial conducted in 2015 [165] failed to demonstrate a pain sensitivity reduction or clinical pain and symptom severity improvement after MC stabilizer (ketotifen) administration. In addition, cytokines’ gene expression, evaluated from skin biopsies of FM patients, reported opposing results: Salemi S. et al. [166] found TNF, IL-1β and IL-6 expression in the skin biopsies of FM patients, while no cytokines could be detected in HC, whereas Üçeyler N. et al. [167] did not find any differences in cytokine expression in the skin samples of FM patients compared to HC. Besides tissue gene expression, different cytokines and chemokines’ levels, either from serum or plasma samples, have been studied extensively, with extremely mixed results. The various results in the literature have been collected in three systematic reviews and meta-analyses [168,169,170].

According to these studies, FM patients present a peripheral blood cytokine profile that differs from that of HC: increased levels of TNF-α [169,170], IL-6 [168,169,170], IL-8 [168,169,170] and IL-10 [167] can be detected. It must be taken into account that cytokine research in FM syndrome remains poorly standardized and several factors—such as the circadian rhythmicity of secretion, environmental factors and medication—could influence cytokine assays and reduce the quality of calculated data [168]. The lack of standardized analytical methods to measure the levels of immune mediators considerably contribute to the increased heterogeneity of the results, thus making it even more difficult to draw any definitive conclusion [170]. However, FM patients may indeed present a particular cytokine signature as a result of complex features, involving potential compensatory anti-inflammatory cytokines and chemokines implicated in the modulation of neuronal plasticity [169]. The actual relation between this cytokine signature and the clinical evidence of pain remains more elusive and appears difficult to establish. Ang D.C. et al. [171] showed a longitudinal relation between chemokine levels, especially IL-8 and monocyte chemotactic protein (MCP-) 1, and the severity of pain, while Wang H. et al. [172] observed a concordant reduction in IL-8 levels and pain intensity during pain therapy but no significant relation was demonstrated, except after six months of treatment. Another plausible hypothesis is that the first cytokine source could be represented by non-neuronal cells in the CNS, rather than peripheral cells. Two studies [173,174] demonstrated IL-8 levels in the CSF of FM patients that were higher than those of HC and rheumatoid arthritis patients; in addition, in FM patients, IL-8 levels in CSF were higher than in peripheral blood samples, leading the authors to conclude that the first source of this chemokine is the CNS—in particular, astrocytes and microglial cells. It has also been hypothesized that thalamic MCs contribute to neuroinflammation and pain by releasing neurosensitizing mediators, which can stimulate diencephalon microglial cells, in the context of central cytokine production [175]. In conclusion, it appears that chronic cytokine production and neuroinflammation are part of the pathophysiology of FM, but different features, such as the existence of a specific cytokine signature, the actual first cytokine source and the clinical relation between cytokines and pain, remain poorly understood.

Thus far, neither peripheral nor CSF cytokine levels can be considered a reliable biomarker of pain in FM syndrome, and additional studies are warranted to increase our knowledge regarding the effective role of inflammation in the pathophysiology of pain in FM patients.

### 3.7. Pentraxin-3

Pentraxin-3 (PTX-3) is an acute-phase glycoprotein, a member of the long pentraxin family, which acts as a modulator of inflammatory processes during microbial or sterile inflammatory responses [176]. PTX-3 synthesis in stimulated by different molecules, such as lipopolysaccharides (LPS), IL-1β and TNF-α, while IL-6 does not influence PTX-3 production [177]. Both protective and detrimental effects of PTX-3 have been described. This glycoprotein has the ability to bind to a variety of microbes, acting as an opsonin and thus stimulating the immune response, but it is also able to protect the organism from infection-induced organ injury and sterile-inflammation-induced tissue damage, limiting excessive immune cell infiltration. Conversely, it has also been described that PTX-3 can worsen tissue injury in certain pathological conditions, such as post-ischemic renal injury, intestinal ischemia and reperfusion or ventilation-induced lung injury [178].

PTX-3 could be considered a plasmatic marker of immune system activation and it has been proposed that FM patients may present augmented PTX-3 levels as a consequence of cytokine dysregulation and increased TNF-α production (Table 5). Skare T.L. et al. [179] demonstrated higher PTX-3 plasmatic levels in FM women compared to HC, in the absence of significant differences between depressed patients and those without depression [180]. In addition, Garcia J.J. et al. [181] found that monocytes from FM women without depression released more PTX-3 than HC, both constitutively and after LPS stimulation, while no difference was found in neutrophils’ ability to release PTX-3. These findings support the hypothesis of increased PTX-3 production in FM patients, suggesting that activated monocytes may be a possible peripheral source, and allow us to include PTX-3 levels as a potential biomarker for FM syndrome. However, further studies are needed to better understand the role of PTX-3 in FM and possibly evaluate whether PTX-3 levels are associated with particular clinical features of FM, especially with clinical evidence of pain.

### 3.8. Neuropeptide Y

Neuropeptide Y (NPY) is a 36-amino-acid peptide widely distributed in the CNS and PNS but also expressed by a variety of immune cells and implicated in many physiological and pathological conditions [182]. NPY preferentially binds two major inhibitory receptors—Y1 and Y2—broadly expressed on the surfaces of different neuronal and non-neuronal cells, leading to the activation of different molecular pathways and consequent biological effects [183]. It is implicated in pain modulation at the DRG and DH levels, where it seems to mediate mostly anti-nociceptive effects. After hyperalgesia-induced excitatory signals, primary afferent neurons and DH inhibitory interneurons upregulate NPY expression and release it in the DH of the spinal cord [184]. Released NPY activates the Y1 and Y2 receptors. The Y1 receptor is expressed on excitatory interneurons localized in the superficial layer—lamina II—of the DH of the spinal cord, and its activation leads to the post-synaptic inhibition of the spinal interneurons and a net reduction in the pain excitatory:inhibitory ratio [185]. Indeed, the Y2 receptor is expressed on the central terminals of small- to medium-sized myelinated A fibers and its activation induces pre-synaptic nociceptive inhibition, especially via pronociceptive substances’—such as SP—reduction [184]. Besides mediating analgesia, NPY activity has also been related to the stress response and resilience-promoting properties. It has been demonstrated that NPY can be expressed in different brain areas, including the hypothalamic arcuate nucleus and excitatory interneurons located in the amygdala [186]. The hypothalamic arcuate nucleus is functionally associated with the paraventricular nucleus of the hypothalamus—the main brain source of CRH—and there is evidence that NPY activity counteracts the biological actions of CRH, favoring stress adaptation processes. On the other hand, NPY released in the amygdala mediates excitatory pyramidal neurons’ inhibition, and the reduced activity of this brain area is associated with anxiolytic effects and increased stress resilience [187]. Moreover, NPY is widely distributed throughout post-ganglionic sympathetic nerves and it is co-stored and co-secreted with noradrenaline, so that it can be considered a marker of noradrenergic function [188]. Lastly, NPY may also act as an immunoregulatory factor, acting on immune system function either directly through NPY receptors expressed on the surfaces of different immune cells or indirectly through the regulation of different physiological or pathological conditions [189].

Considering the known relationship between stressful stimuli exposure and FM symptom onset, as well as the evidence of a maladaptive response to these stimuli, authors have investigated the possible application of NPY peripheral levels as a biomarker of ANS dysfunction and its potential association with pain experience (Table 6). Different studies describe not only reduced plasma levels of NPY in FM patients compared to HC [190,191] but also low NPY levels after 30 min on a tilt table [191,192], thus suggesting the hypofunction as well as hyporeactivity of the sympathetic stress axis in FM patients. Conversely, subsequent studies demonstrated higher NPY plasma and serum levels in FM patients compared to HC [188,193,194] and lower HRV in FM patients, also showing a significant relation between HRV and NPY [193]. These findings suggest that FM patients could develop increased sympathetic–adrenal activity, possibly as a maladaptive response to high stress or chronic tension. In none of these studies [188,193,194] was a significant relationship between NPY levels and clinical evidence of pain present. These findings suggest the contribution of impaired ANS activation and altered NPY expression to the pathophysiology of FM, but the exact role of sympathetic dysfunction remains less understood, mostly due to the complexity of this system and its implications in a variety of biological conditions. To date, there is no evidence of a direct association of NPY levels with any FM symptoms; thus, so far, NPY cannot be considered a useful biomarker of pain in FM syndrome.

## 4. Potential Future Directions

### 4.1. Vitamin D

Recently, the possible involvement of hypovitaminosis D in the pathophysiology of chronic pain disorders has elicited increasing interest among the scientific community. Vitamin D, beyond its fundamental role in skeletal and calcium homeostasis, has been recognized to be implicated in a variety of other actions, including a potential regulatory function in pain pathways [195]. On a clinical level, vitamin D deficiency is associated with osteomalacia, a pathological condition characterized by the development of bone pain, proximal muscle weakness and generalized fatigue and commonly misdiagnosed as FM syndrome or other chronic pain conditions [196]. In addition, both vitamin D receptor (VDR) and 1α-hydroxylase—the enzyme that converts 25-hydroxyvitamin D (25(OH)D) by hydroxylation into the active 1,25 di-hydroxyvitamin D (1,25(OH)2D3)—have been found in the DRG, spinal cord and different pain-processing brain areas, expressed either by neuronal or glial cells [197], but the underlying molecular mechanisms by which vitamin D/VDR may modulate pain pathways remain scarcely understood [198]. However, a large number of studies have been conducted to explore the potential association of low levels of serum vitamin D with pain in FM syndrome. Different systematic reviews of the literature have been conducted, focusing on the possible association between vitamin D and FM, and, considering the highest-quality available studies, it would appear that FM patients are characterized by lower vitamin D levels than HC [199]. According to a meta-analysis conducted by Makrani A.H. et al. [200], FM patients present significantly lower vitamin D levels compared to HC, and a second meta-analysis [201] found a positive association between low serum vitamin D levels and chronic widespread pain conditions, including FM. However, since these results came from observational studies, no causal relationship can be inferred. Conversely, a third meta-analysis [202] did not find any difference in the circulating concentration of vitamin D or vitamin D deficiency between FM patients and HC. According to the authors, most of the considered studies presented poor-quality data and considerable heterogeneity, making it difficult to reach accurate final conclusions regarding the vitamin D status in FM patients. Unfortunately, the majority of the available studies present low statistical power and several types of bias, so that current evidence regarding an association between FM syndrome and vitamin D deficiency and their cause–effect relationship remains inconclusive [199,203,204]. In addition, methodological issues regarding vitamin D assays exist. First, different methodological approaches exist, with extreme variability in precision among them; secondly, there is still no consensus regarding the ideal serum level of vitamin D and, consequently, the most appropriate cut-off to define hypovitaminosis D in the general population [203].

To date, extremely discordant data exist regarding the potential role of hypovitaminosis D in FM patients and, even if a higher prevalence of hypovitaminosis D cannot be excluded in these patients, questions remain regarding the effective pathophysiological role of vitamin D. Possible alternative explanations are that hypovitaminosis D could be a consequence of limited exercise and sun exposure, both favored by chronic pain [205], or it could simply reflect a characteristic of the general local population considered [203]. Future research is warranted, focusing on prospective study designs that exhaustively account for confounders and potentially problematic methodological pitfalls, in order to ascertain any causative role of vitamin D in the development of FM [199]. In a similar way, evidence regarding the effect of the supplementation of vitamin D on FM symptoms, including pain and quality of life, remains inconclusive [204]. Safe clinical conclusions are hindered by several limitations of the studies conducted, such as the absence of a control group, small sample size and discrepancies in the type, regimen and dose of vitamin D supplements administered [197]. Nonetheless, authors agree that FM patients should be screened for hypovitaminosis D and, if it is present, supplementation should be administered not only to maintain bone health [204] but also for the possible positive impact on pain perception and quality of life [205].

### 4.2. Gut Microbiome

The gut microbiome is the most complex and populous micro-ecological system in the body, and it is highly variable among healthy individuals due to the influence of many factors, including genetic, physiological, psychological and environmental determinants [206]. It is now clear that homeostasis between the gut microbiome and the host is essential for the maintenance of a healthy status, given the regulating role of the gut microbiome in a variety of different conditions, such as metabolic pathways, gut barrier integrity, protection from pathogens, brain development and immune system function [207]. In addition, increasing evidence exists of bidirectional communication between the gut and brain, and the gut microbiome is nowadays considered a key gastrointestinal regulatory factor in the so-called gut–brain axis, thus participating in the development of different neurological disorders, including chronic pain syndromes [208]. The gastrointestinal microbiota can directly or indirectly modulate the peripheral sensitization underlying chronic pain through multiple gut-microbiota-derived mediators, including pathogen-associated molecular patterns (PAMPs), metabolites—such as short-chain fatty acids and bile acids—and neurotransmitters or neuromodulators’ release. It has also been proposed that the gut microbiome could be implicated in the direct and indirect regulation of neuroinflammation-mediated CS [209]. Recently, it has been suggested that changes in the gut microbiome could play a role in the pathogenesis of FM and associated symptoms’ severity. Clos-Garcia M. et al. [210] first described differences in the gut microbiome compositions of FM patients and HC, also identifying correlations with different serum metabolite alterations in FM patients. A subsequent interesting review [211] highlighted that some of the species identified as differentially abundant are microorganisms characterized by metabolic activity, especially short-chain-fatty-acid-producing bacteria and species implicated in bile acid metabolism, suggesting a possible correlation of these species with the clinical phenotype and symptom severity in FM patients. Thus, it appears that an association between the composition and metabolism of the gut microbiome and FM symptoms may exist. However, it must be taken into account that current data are limited, characterized by the variance of findings, heterogeneity of the techniques applied and the possible influence of confounding factors [212]. Therefore, further well-designed studies are warranted in order to confirm the possible association between the gut microbiome and FM syndrome and symptoms and to hopefully identify more specific possible biomarkers for the disease.

## 5. Materials and Methods

A bibliographic search strategy was conducted to identify all studies reporting metabolites possibly associated with pain in FM. An electronic database search of PubMed was performed. First, we searched for studies that investigated pathogenetic mechanisms associated with pain in FM, in order to define the most relevant metabolites to include. We first used key terms that included ‘fibromyalgia’, ‘pain’, ‘central sensitization’, ‘neuroinflammation’, ‘hypothalamic-pituitary-adrenal axis disfunction’. For specific biomarkers, we subsequently applied the following search: ‘fibromyalgia pain’ AND ‘glutamate’; ‘fibromyalgia pain’ AND ‘substance P’; ‘fibromyalgia pain’ AND ‘nerve growth factor’; ‘fibromyalgia pain’ AND ‘brain derived neurotrophic factor’; ‘fibromyalgia pain’ AND ‘mu opioid receptor’; ‘fibromyalgia pain’ AND ‘mast cells’ OR ‘cytokine levels’; ‘fibromyalgia pain’ AND ‘pentraxin-3′; ‘fibromyalgia pain’ AND ‘neuropeptide Y’; ‘fibromyalgia pain’ AND ‘vitamin D’; ‘fibromyalgia pain’ AND ‘gut microbiome composition’ OR ‘gut microbiome metabolite’. Only research performed in humans was considered, while studies conducted on animal models were excluded. Only full-text articles in English were considered. Finally, after reading abstracts for relevance, 67 papers were included. The process of the selection of the studies can be found in Figure 1.

## 6. Conclusions

In this narrative review, we summarized the actual knowledge regarding metabolites associated with chronic pain development, whose alteration has been linked to FM syndrome. We not only considered more extensively studied metabolites but also possible new areas of research. Among them, more consistent evidence is available regarding Glu, SP, BDNF and NPY level alterations. In particular, dysfunction in Glu metabolism is apparently implicated in chronic pain development in FM and, therefore, Glu level detection might be considered useful in these patients. Unfortunately, the strongest evidence is related to Glu level assessment in particular brain regions using functional neuroimaging techniques, which, currently, are difficult to apply in clinical practice. On the other hand, the altered production of SP, BDNF and NPY seems reasonably implicated in FM pathophysiology but the association of these metabolites with the clinical evidence of pain is still unclear. It must be noted that there is a large discrepancy between studies regarding the inclusion and exclusion criteria used, sources of samples considered and tests used to assess metabolite levels, thus making it even more difficult to compare results. Based on the results of the analyzed studies, it seems unlikely that one of these metabolites could be used as a biomarker for FM diagnosis in the near future. Research in this field has, however, been important since it has contributed to clarifying at least part of the pathogenetic mechanism underlying this complex syndrome.

Besides diagnostic porpoises, the identification of a biomarker for FM might be considered useful in the monitoring of treatment efficacy. As a matter of fact, some of the metabolites included in this review—in particular, SP, BDNF and cytokine levels—have been used to monitor different treatment strategies, reporting conflicting results. In our opinion, since there is still little evidence regarding the association of these markers with any clinical symptoms—especially pain—their actual usefulness in monitoring treatment efficacy remains unclear. Thus, more studies in this area should be conducted.

In summary, many research areas are currently under investigation for the identification of possible biomarkers for FM syndrome. These studies also enable us to deepen our knowledge regarding the pathophysiological mechanisms underlying this heterogenous condition. Since WP is the major symptom in FM patients, we decided to focus on biomarkers under investigation for their possible associations with the clinical evidence of pain. Thus far, none of the proposed metabolites is sufficiently reliable to be validated as a diagnostic biomarker, and the different methodological issues mostly remain unresolved. Considering the clinical complexity and variability of this syndrome, we can conclude that a single biomarker is not sufficiently reliable; however, in the future, a panel of biomarkers, including subtype-specific biomarkers, could aid in FM diagnosis.

## Figures and Tables

**Figure 1 ijms-24-10443-f001:**
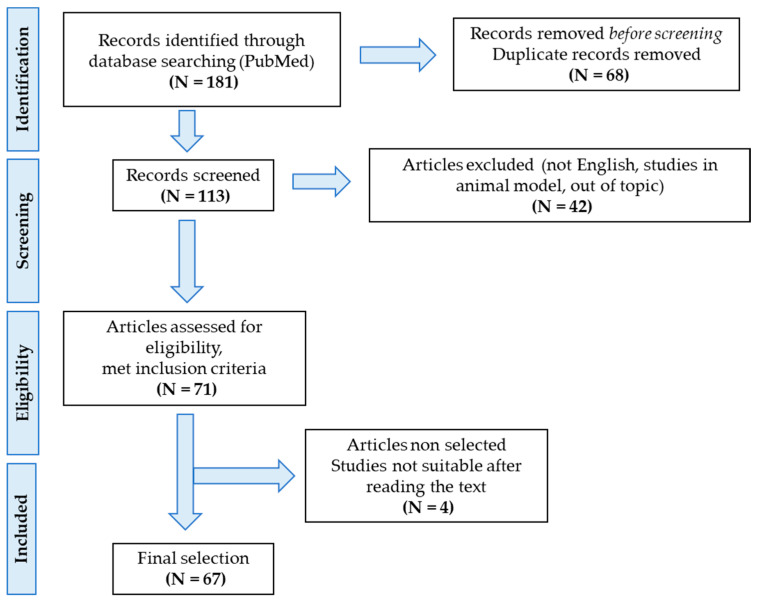
Flow chart showing the study selection process.

**Table 1 ijms-24-10443-t001:** Glutamate levels and association with pain in fibromyalgia patients.

	Population Sample	Specimen	Results	Correlation with Pain
Peres MF et al., 2004 [74]	8 CM pts, 12 FM pts and 20 HC	CSF Glu levels measured by HPLC	Higher Glu levels in FM than CM (*p* < 0.04) and HC (*p* < 0.001)	Positive correlation of mean pain scores with Glu levels (*r* 0.551 *p* < 0.012)
Sarchielli P et al., 2007 [75]	20 FM pts, 20 CM pts and 20 HC	CSF Glu levels measured by HPLC	Higher Glu levels in FM (*p* < 0.003) and CM (*p* < 0.001) than HC	Absence of correlation between Glu levels and VAS values and pain intensity, pressure pain threshold and TPC
Harris RE et al., 2009 [76]	19 FM pts and 14 HC	Posterior right insula Glu levels assessed by H-MRS	Higher Glu levels in FM pts (*p* < 0.009) than HC	Inverse correlation of low (*r* −0.53 *p* < 0.002) medium (*r* −0.43 *p* < 0.012) and high (*r* −0.38 *p* < 0.03) pressure pain thresholds with posterior insula Glu levels
Fayed N et al., 2010 [78]	10 FM pts and 10 HC	Posterior cingulate gyrus Glx and Glx/Cr levels assessed by MRS	Higher Glx (*p* < 0.049) and Glx/Cr levels in FM pts (*p* < 0.034) than HC	Inverse correlation of pain threshold assessed by sphygmomanometer with posterior cingulate Glx (*r* −0.45 *p* < 0.047) and Glx/Cr levels (*r* −0.50 *p* < 0.024)
Valdés et al., 2010 [79]	28 FM pts and 24 HC	Right amygdala Glx and Glx/Cr levels assessed by H-MRS	Higher Glx (*p* < 0.03) and Glx/Cr levels in FM pts (*p* < 0.04) than HC	Absence of correlation with pain
Feraco T et al., 2011 [80]	12 FM pts and 12 HC	Right VLPC and left thalamus Glu/Cr levels assessed by MRS	Higher Glu/Cr levels in FM pts (*p* < 0.01) than HC	Positive correlation of VAS pain (*r* 0.73 *p* < 0.007) with left thalamus Glu/Cr levels

CM, chronic migraine; pts, patients; FM, fibromyalgia; HC, healthy control; CSF, cerebrospinal fluid; Glu, glutamate; HPLC, high-performance liquid chromatography; VAS, visual analog scale; TPC, tender point count; ^1^H-MRS, magnetic resonance spectroscopy; Glx, glutamate + glutamine; Cr, creatinine.

**Table 2 ijms-24-10443-t002:** Substance P levels and association with pain in fibromyalgia patients.

	Population Sample	Specimen	Results	Correlation with Pain
Vaerøy H, et al., 1988 [92]	30 FM pts	CSF SP levels measured by RIA technique	Higher SP levels in FM pts (*p <* 0.001) compared to normal value	Not evaluated
Reynolds WJ et al., 1988 [95]	32 FM pts and 26 HC	Plasma SP levels measured by RIA technique	No significant difference between FM pts and HC	Not evaluated
Russel IJ et al., 1994 [93]	32 FM pts and 30 HC	CSF SP levels measured by RIA technique	Higher SP levels in FM pts (*p <* 0.001) than HC	Absence of correlation between SP levels and pain
Tsilioni I et al., 2016 [96]	84 FM pts and 20 HC	Serum SP levels measured by ELISA technique	Higher SP levels in FM pts (*p <* 0.0001) than HC	Not evaluated

FM, fibromyalgia; pts, patients; CSF, cerebrospinal fluid; SP, substance P; HC, healthy control; RIA, radioimmunoassay; ELISA, enzyme-linked immunosorbent assay.

**Table 3 ijms-24-10443-t003:** Nerve growth factor levels and association with pain in fibromyalgia patients.

	Population Sample	Specimen	Results	Correlation with Pain
Sarchielli P et al., 2007 [75]	20 FM patients, 20 CM patients and 20 HC	CSF NGF levels measured by ELISA technique	Higher NGF levels in FM pts (*p <* 0.001) and CM pts (*p <* 0.0005) than HC	Positive correlation of NGF levels with duration of chronic pain (*r* 0.66 *p <* 0.003); absence of correlation between NGF levels and VAS values and pain intensity, pressure pain threshold and TPC
Baumeister D et al., 2019 [120]	97 FM patients and 35 HC	Serum NGF levels measured by ECL assay technique	No significant difference between FM pts and HC	Not evaluated
Jablochkova A et al., 2019 [121]	75 FM patients and 25 HC	Plasma NGF levels measured by ECL assay technique	Lower NGF levels in FM pts (*p <* 0.001) than HC	Absence of correlation between NGF levels and global pain intensity, pressure pain threshold and TPC

FM, fibromyalgia; CM, chronic migraine; HC, healthy control; CSF, cerebrospinal fluid; NGF, nerve growth factor; ELISA, enzyme-linked immunosorbent assay; VAS, visual analog scale; TPC, tender point count; ECL, electrochemiluminescence.

**Table 4 ijms-24-10443-t004:** Brain-derived neurotrophic factor levels and association with pain in fibromyalgia patients.

	Population Sample	Specimen	Results	Correlation with Pain
Sarchielli P et al., 2007 [75]	20 FM pts, 20 CM pts and 20 HC	CSF BDNF levels measured by ELISA technique	Higher BDNF levels in FM pts (*p <* 0.001) and CM pts (*p <* 0.0001) than HC	Positive correlation of BDNF levels with duration of chronic pain (*r* 0.57 *p <* 0.01); absence of correlation between BDNF levels and VAS values and pain intensity, pressure pain threshold and TPC
Laske C et al., 2007 [132]	41 FM pts and 45 HC	Plasma BDNF levels measured by ELISA technique	Higher BDNF levels in FM pts (*p <* 0.0001) than HC	Not evaluated
Haas L et al., 2010 [131]	30 FM pts and 30 HC	Plasma BDNF levels measured by ELISA technique	Higher BDNF levels in FM pts (*p <* 0.049) than HC	Absence of correlation between BDNF levels and VAS values and TPC
Nugraha B et al., 2013 [134]	28 FM pts and 27 HC	Serum BDNF levels measured by ELISA technique	Higher BDNF levels in FM pts (*p <* 0.05) than HC	Absence of correlation between BDNF levels and pain and TPC
Ranzolin A et al., 2016 [137]	69 FM pts and 61 HC	Serum BDNF levels measured by ELISA technique	No significant difference between FM pts and HC	Not evaluated
Baumeister D et al., 2019 [121]	97 FM pts and 35 HC	Serum BDNF levels measured by ELISA technique	No significant difference between FM pts and HC	Not evaluated
Jablochkova A et al., 2019 [121]	75 FM pts and 25 HC	Plasma BDNF levels measured by ECL assay technique	Higher BDNF levels in FM pts (*p <* 0.001) than HC	Absence of correlation between BDNF levels and global pain intensity, pressure pain threshold and TPC
Iannuccelli C et al., 2022 [138]	40 FM pts and 40 HC	Serum BDNF levels measured by ELISA technique	Lower BDNF levels in FM pts (*p <* 0.0001) than in HC	Absence of correlation between BDNF levels and TPC
Bidari A et al., 2022 [143]	53 FM pts and 23 non-FM chronic nociceptive pain pts	Serum BDNF levels measured by ELISA technique	No significant difference between FM pts and non-FM chronic nociceptive pain pts	Negative correlation of BDNF levels with VAS pain (*r* −0.32, *p* < 0.05)

FM, fibromyalgia; pts, patients; CM, chronic migraine; HC, healthy control; CSF, cerebrospinal fluid; BDNF, brain-derived neurotrophic factor; ELISA, enzyme-linked immunosorbent assay; VAS, visual analog scale; TPC, tender point count; ECL, electrochemiluminescence.

**Table 5 ijms-24-10443-t005:** Pentraxin-3 levels and association with pain in fibromyalgia patients.

	Population Sample	Specimen	Results	Correlation with Pain
Skare TL et al., 2015 [176]	94 FM pts and 94 HC	Plasma PTX-3 levels measured by ELISA technique	Higher PTX-3 levels in FM pts (*p <* 0.005) than HC	Absence of correlation between PTX-3 levels and pain
Garcia JJ et al., 2016 [178]	10 FM pts and 10 HC	PTX-3 release by phagocytes, measured by ELISA technique	Higher PTX-3 constitutive release in FM pts (*p <* 0.05) than HC	Not evaluated

FM, fibromyalgia; pts, patients; HC, healthy control; PTX-3, pentraxin-3; ELISA, enzyme-linked immunosorbent assay.

**Table 6 ijms-24-10443-t006:** Neuropeptide Y levels and association with pain in fibromyalgia patients.

	Population Sample	Specimen	Results	Correlation with Pain
Crofford LJ et al., 1994 [187]	12 FM pts and 10 HC	Plasma NPY levels measured by RIA technique	Lower NPY levels in FM pts (*p <* 0.001) than HC	Not evaluated
Anderberg UM et al., 1999 [191]	24 FM pts and 17 HC	Plasma NPY levels measured by RIA technique	Higher NPY levels in FM pts (*p <* 0.002) than HC	Absence of correlation between NPY levels and pain
Di Franco M et al., 2009 [190]	51 FM pts, 25 SSc pts and 15 HC	Serum NPY levels measured by immunoenzymatic assay technique	Higher NPY levels in FM pts (*p <* 0.0001) than SSc patients and HC	Absence of correlation between NPY levels and pain
Iannuccelli C et al., 2010 [185]	51 FM pts, 25 TTH patients and 15 HC	Serum NPY levels measured by immunoenzymatic assay technique	Higher NPY levels in FM pts (*p <* 0.0001) than HC	Absence of correlation between NPY levels and pain

FM, fibromyalgia; pts, patients; TTH, tension-type headache; HC, healthy control; NPY, neuropeptide Y; RIA, radioimmunoassay; SSc, systemic sclerosis.

## Data Availability

Not applicable.

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
