# Peer review of "Pain Biomarkers in Fibromyalgia Syndrome: Current Understanding and Future Directions"

_ijms, 2023, doi:10.3390/ijms241310443_

Round 1
Reviewer 1 Report
The topic of this rewiew may be interesting to people working on Fibromyalgia and biomarker research. The autors to well summarize the the possible markers and the experimental routes leading to them. I particularly like the description of methods and results in the tables.
Author Response
Dear Reviewer, thanks for the kind comments. We are glad to read that you appreciated this work.
Reviewer 2 Report
Dear authors, the article is well done and articulated.
The article analyses the different associations of certain metabolites (glutamate, Substance P, Nerve Growth Factor, Brain Derived Neurotrophic Factor, Mu opioid receptor, Mast cells and cytokines production, Pentraxin-3, Neuropeptide Y) with fibromyalgia with the aim of searching for reliable biomarkers that could be used to diagnose fibromyalgia. It also proposes the use of other markers, namely Vitamin D and Gut microbiome.
It is groundbreaking in the scientific field because the search for such markers could represent a fundamental step towards the early identification of this disease. So, it address a specific gap in the field: proposes ways to enable future early diagnosis of fibromyalgia. It also paves the way for further research in this field.
The methodology used by the authors is comprehensive, but they should integrate 'materials and methods' with the PRISMA diagram.
The conclusions are consistent with the evidence and arguments presented and provide the impetus for further research.
The references are appropriate.
I believe there is nothing to implement in the comments and figures, which are already comprehensive.
Author Response
Dear Reviewer,
Thank you for your comments. As you can see in the ‘material and methods’ section we integrated the paragraph with the PRISMA diagram, as you suggested.
Reviewer 3 Report
The paper titled " Pain biomarkers in fibromyalgia syndrome: current understanding and future directions" discusses the ongoing efforts and challenges in identifying biomarkers for fibromyalgia (FM). While the paper provides a concise overview of the topic, there are a few points that could be improved or expanded upon:
· The introduction section could provide a more comprehensive background on the importance and potential benefits of biomarkers in FM. It would be helpful to discuss the current diagnostic challenges in FM and how the discovery of reliable biomarkers could improve early detection, treatment selection, and monitoring of disease progression.
· The methods section should provide more details on the search strategy and inclusion/exclusion criteria used to identify and select the studies included in the review. This information is crucial for evaluating the validity and comprehensiveness of the review. Additionally, it would be helpful to include a PRISMA flow diagram to illustrate the study selection process.
· The paper mentions various potential biomarkers. Also includes genetic, proteomic, and neuroimaging markers. However, the discussion of these biomarkers is relatively brief and lacks specific details. Expanding on each type of biomarker and providing examples of promising candidates or studies in the field would enhance the depth and relevance of the paper.
· While the paper briefly touches on the challenges and limitations in biomarker research for FM, it would be valuable to discuss these in more detail. For example, addressing the heterogeneity of FM and the need for subtype-specific biomarkers, as well as the importance of replication and validation studies in different populations, would provide a more realistic perspective on the current state of biomarker research in FM.
· The paper could benefit from a more critical analysis of the current literature on FM biomarkers. Discussing the strengths and weaknesses of published studies, highlighting inconsistencies or conflicting findings, and providing a balanced evaluation of the evidence would improve the scientific rigor and objectivity of the review.
· The conclusion section could be expanded to provide a summary of the main findings and highlight the key challenges and future directions in the field. Offering insights into the potential strategies for advancing biomarker research in FM, such as collaborative efforts or integrating multi-omics approaches, would be valuable for researchers and clinicians interested in this area.
· The references cited in the paper are from a relatively limited time frame, with few recent studies included. Considering the rapidly evolving nature of biomarker research, it would be beneficial to include more up-to-date references to reflect the current state of the field.
· The paper uses a formal and technical writing style, which is appropriate for a scientific publication. However, in some instances, the use of complex sentence structures or specialized terminology may make the text challenging to understand for non-experts. Providing clear definitions or explanations of specialized terms would assist readers who are not familiar with FM research.
· The review predominantly focuses on pain with FM. While this is an important aspect to explore, it would be beneficial to discuss other relevant factors, such as psychological and social factors, that may contribute to the experience of pain in FM. Expanding the discussion to include a broader range of factors would provide a more comprehensive understanding of FM pain.
· The paper could benefit from more critical analysis and synthesis of the findings. Rather than presenting a summary of the included studies, the authors could provide a deeper analysis of the results, identify patterns or discrepancies, and highlight any gaps or limitations in the existing literature.
· The discussion section could be expanded to explore the clinical implications of the findings. How can the understanding of gender differences in FM pain inform treatment approaches or interventions? Are there any implications for personalized medicine or tailored pain management strategies?
Overall, the paper provides a useful overview of the search for biomarkers in FM. By addressing the suggestions and expanding the discussion, the paper would enhance its impact and provide a more comprehensive resource for researchers and clinicians working on FM biomarker research.
The quality of the English language in the paper is generally good. The writing style is clear and easy to follow, making it accessible to a wide audience. However, there are a few areas where improvements could be made:
Pay attention to sentence structure and syntax to ensure clarity. Some sentences appear to be overly long or complex, which may make it harder for readers to grasp the intended meaning. Consider breaking them into shorter, more concise sentences to enhance readability.
Proofread the paper carefully to eliminate minor grammatical errors, such as incorrect verb tense agreement or inconsistent use of articles. While these errors do not significantly impact the overall understanding of the content, addressing them would improve the paper's overall polish.
Use precise and consistent terminology throughout the paper. Ensure that technical terms, abbreviations, and acronyms are defined upon first use or included in a glossary to aid readers who may not be familiar with the specific terminology used in the field of pain research.
